

# Brief Communication: The Danish Replicate Drilling System – Results from the First Field Test

Julien Westhoff[1], Grant Vernon Boeckmann[1], Nicholas Mossor Rathmann[1], and Steffen Bo Hansen[1]

[1]Niels Bohr Institute, University of Copenhagen, 2200 Copenhagen, Denmark

*Correspondence to*: Grant Boeckmann (grant.boeckmann@nbi.ku.dk)

**Abstract.**

We report on the successful test of a new replicate drilling system for ice cores. The test was done in drill fluid, at 140 m depth of the EastGRIP borehole in central Greenland. By determining the borehole orientation, broaching into the side of the borehole wall, we can guide the milling tool into the downhill side of the borehole. Thus, we produce a ledge on which we rest the drill with all its weight. Gravity would now guide the ice core drill into the downhill side of the hole, gradually producing full ice cores.


## 1 Introduction

### 1.1 EastGRIP and future projects

The East Greenland Ice Core Project (EastGRIP or EGRIP) drilling finished in 2023 by reaching the subglacial environment, i.e. a water-saturated mud layer, at a depth of approx. 2666.7 m. The 2024 field season was leveraged to test bedrock and

replicate/directional drilling tools in the borehole while the camp was being prepared to traverse. In late May 2024, we tested a three-stage replicate drilling system in the upper (140 m depth) and liquid filed part of the EastGRIP borehole. The aim was to accomplish a first field test with this newly designed system. In the future, the system is planned to be used at the GRIP site in 2026, with the aim of redrilling the basal and bottom-most layers of the GRIP borehole, close to the summit of the Greenland ice sheet (GRIP members, 1993). Furthermore, the replicate system is intended to be used at the Little Dome C site for the

*"Beyond European Project for Ice Coring in Antarctica Oldest Ice"* (BEOI) project to retrieve a second ice core from 700 kyr to 1500 kyr b2k period at a depth of around 2500m (Chung et al., 2024).



## 1.2 Previous attempts for replicate drilling

A simple and effective method to produce a replicate ice core is to use the whipstock approach (e.g. Vasiliev et al., 2007).
Herby a slanting device is lowered into the borehole, sat on the bottom, or anchored in the borehole wall, which guides the drill into the side of the borehole to produce a replicate core. This is the so-called passive replicate coring. The method has proven to work well in e.g. the Russian Vostok ice core drilling in Antarctica (Vasiliev et al., 2007). The whipstock method for electromechanical drills is discussed by Shturmakov et al. (2014) and compared to the active replicate coring system, designed to fit the US Deep Ice Sheet Coring (DISC) Drill.

During the North Greenland Eemian Ice Drilling (NEEM) a replicate core was drilled while removing drill chips from the bottom of the borehole (Popp et al., 2014). This occurred due to the switch from the short to long drill, which could not follow the small curvature of the hole and therefore deviated.

Another test was performed at the NEEM site in a dry 400 m deep hole. The borehole had an inclination of around four degrees. Drilling into the side of the borehole was successfully done using only the force of gravity to mill the side over many hours.
A ledge was formed and the first step for a replicate borehole was initiated. The results from this test have not been published but Steffen Bo Hansen has taken part in these tests and in the work presented in this paper.

## 1.3 Initial idea from the inclination correction tool

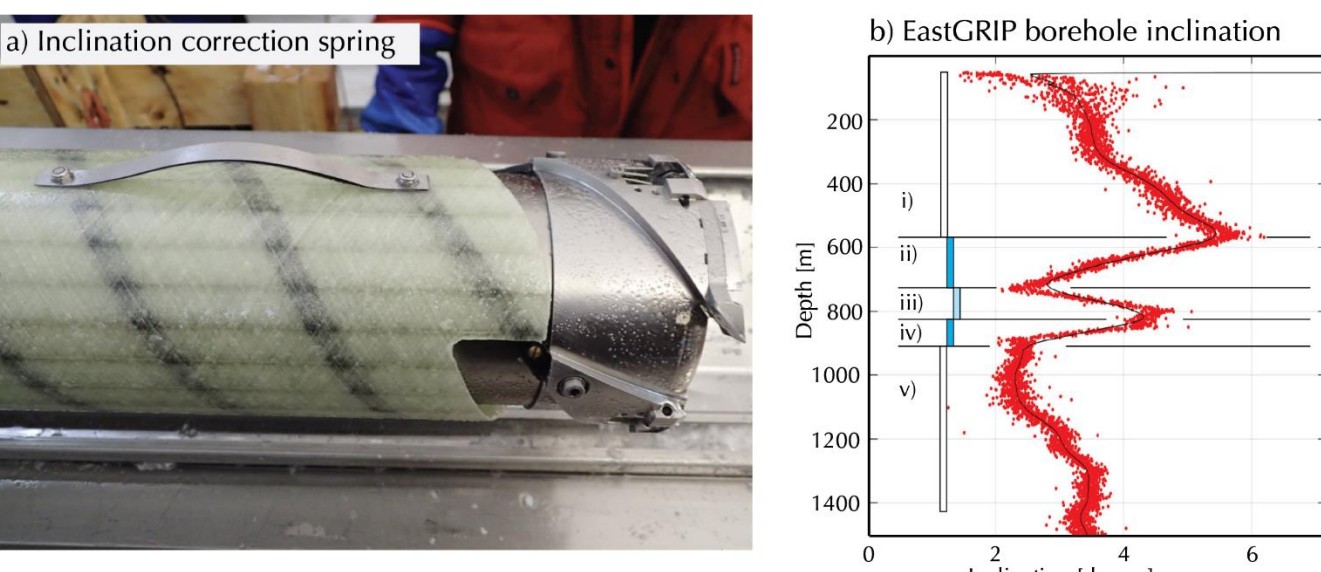

**Fig. 1: a) The inclination correction tool, a spring attached to the outer barrel, designed on-site to decrease borehole inclination at**
**the EastGRIP in 2018. b) Borehole inclination of the upper 1500 m of the EastGRIP hole (pers. comm. Dorthe Dahl-Jensen). Section i) To increase the descent speed of the drill, a deadweight was added to the lightweight short version of the deep drill. This caused the drill's center of gravity to be above the supporting knobs, leading to a rise in the inclination when drilling. ii) We mounted the inclination spring and repositioned the support knobs, gradually decreasing the inclination. iii) We mounted the spring the wrong way around and the inclination increased again. iv) We mounted the spring in the right way again. v) We removed the spring**
**as the inclination was now acceptable.**



In 2017 and 2018, we unintentionally drilled with high inclinations in the upper sections of the EastGRIP borehole (Fig. 1b, section i). Due to the slow descent speed of our lightweight short version of the deep drill, we added a deadweight. This shifted the drill's center of gravity upwards, above the supporting knobs, leading to a rise in inclination while drilling.

To correct the high borehole inclination, we added a spring to the outer barrel (fig. 1), which pushed on the uphill side of the

borehole - forcing the cutterhead toward plumb, thus gradually decreasing the inclination (fig. 1b, section ii). The successful implementation of this technique verified the use of our orientation package, based on the BOSCH BNO055 9-axial orientation sensor, and that slight spring pressure can slowly change the direction of the drill with depth.

We then, accidentally, mounted the spring the wrong way around, increasing the inclination of the borehole once again (fig. 1b, section iii). Once we noticed this, we again mounted the spring in the correct orientation (fig. 1b, section iv). At a depth of

approx. 900 m the borehole inclination was acceptable and we removed the spring (fig. 1b, section v). The borehole inclination data are from pers. comm. with Dorthe Dahl-Jensen and can be found in Supplement 8.

The idea for our directional drilling design is based on this inclination correction tool (fig. 1) and the gravitation-based milling test in the 400 m deep dry hole at NEEM (mentioned above).





## 2 The Replicate Drilling System

### 2.1 Concept and Design Requirements

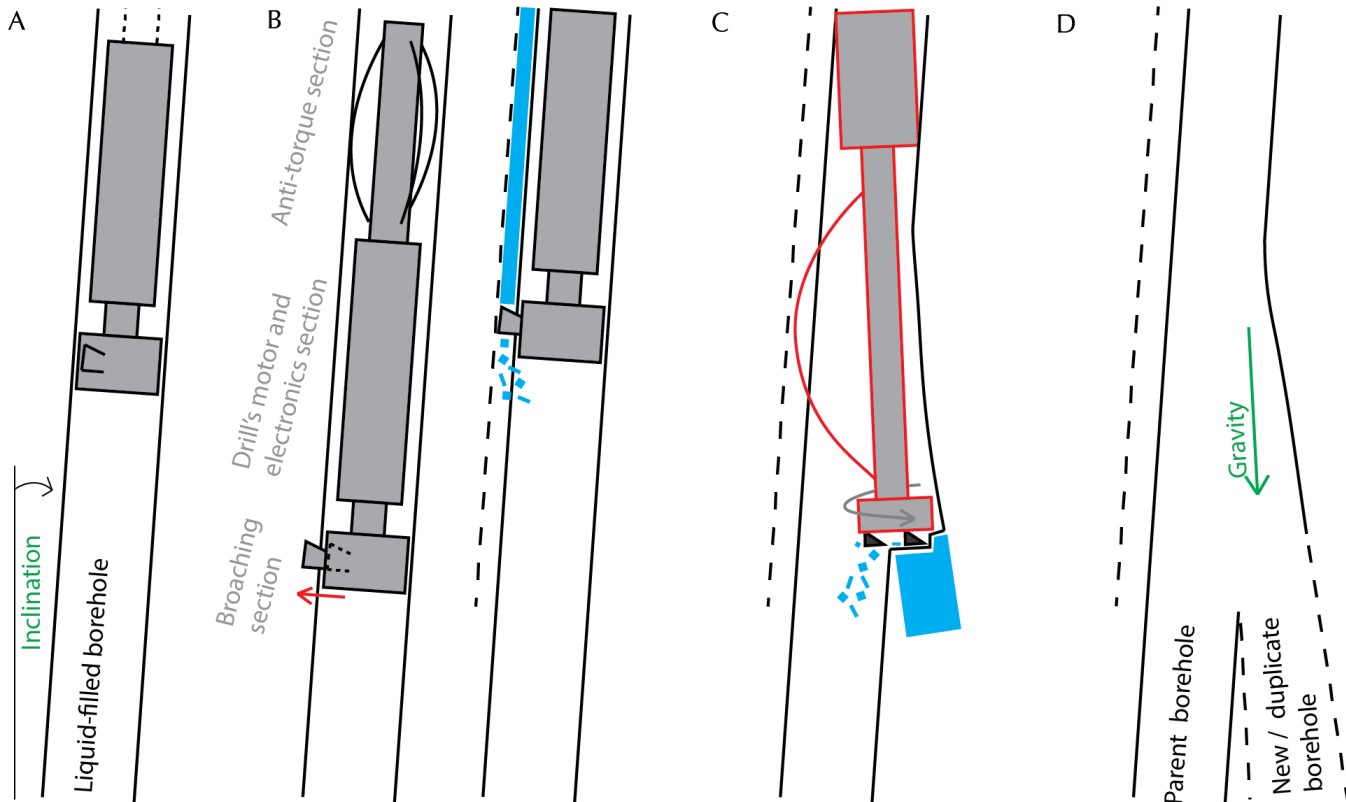

**Figure 2: Concept of the directional drilling setup. A) the broaching tool is lowered into the liquid-filled and slightly inclined borehole. B) The cutter is engaged into the borehole wall (red arrow), and then the tool is pulled up, cutting a groove into the side of the wall. C) The broaching tool is replaced by the spring sleeve and the spring follows the groove (dashed line). The milling head cuts into the side of the borehole wall. D) When lowering the ice core drill, gravity will guide it into the new borehole. In our test we did not perform step D, but stopped at step C.**

We tested the replicate ice coring system in an inclined and liquid-filled borehole (fig. 2a), yet both are not requirements for our system. The concept of the system is the following: first, we determine the spatial orientation of the borehole, then broach the "uphill" side of the hole (fig. 2b), then mill the side of the borehole as shown in fig. 2c. Gravity will then guide downhole tools into the new hole (fig. 2d).

Furthermore, we are limited by some design requirements:

1) the parent hole should remain accessible. It is acceptable to deviate into the downhole side of the parent borehole. This means, that gravity will guide the drill and other downhole tools into the new borehole, but access to the parent hole remains.

2) The system should work in all holes with inclinations between 0° (plumb) and any inclination that can be drilled with a Hans Tausen (HT) style drill.



3) The system should only use the HT Deep Drill pressure tube and hardware for actuation and communication, i.e. the replicate hole and parent hole must have the same diameter, and

4) the system must be operational in temperatures down to -55°C, e.g. for the BEOI project.

5) Deviation from the parent borehole must be possible at any depth (below the casing).



## 2.2 Overview of the full Process



**Figure 3: a, b) Overview of the orientation, broaching, milling, and verification run. c, d) show the broaching procedure in detail, including the stick-slip of the broaching tool and the resulting cable tension, e, f) the procedure of the milling process. g-j) show the plots of the verification of having cut a ledge, by reducing the cable tension by resting the drill on the newly cut ledge.**

On May 27, 2024, we tested orientation around 14:30 and just prior to broaching we set the orientation of the broaching tool (fig. 3a). We broached around 15:00. We used May 28 to clean the borehole from the chips created by broaching the side of





the hole. May 29 and 30 were dedicated to other downhole activities, non-related to replicate drilling. On May 31, 2024, we
first tested the process of slow milling in the morning (not plotted) and then at "normal"-speed (milling, from 15:15 to 15:45).
We verified that we had produced a ledge, just after milling, until 16:05 (fig. 3b). The details will be discussed in the following
sections.

**2.3 Spatial Borehole Orientation**

To ensure we deviate the drill in the intended direction we need to determine the spatial orientation of the borehole inclination,
relative to the drill (fig. 2a and the timing of finding the spatial orientation is indicated in fig. 3a and c). For deviation drilling,
we will mill the "downhill" side of the borehole, so the drill glides into the "new" hole by gravity. However, for this field test,
we attempted to deviate towards the top side of the borehole, to prevent damage that could limit access to the bottom parts of
the hole for future loggings or basal measurements.
To determine the correct position of the broacher, i.e. on the bottom side of the borehole, we used the afore-mentioned drill
orientation package included in the Danish deep drill system, relying on the proprietary "BOSCH Sensor Fusion" software to
calculate the BNO055 orientation quaternion, included in the Danish deep drill system. This software supplies the inclination,
azimuth, and roll of the drill. For our test, azimuth and roll were offset by 180 degrees, meaning the orientation-roll of the drill
is on the opposite side of the direction of maximum inclination.






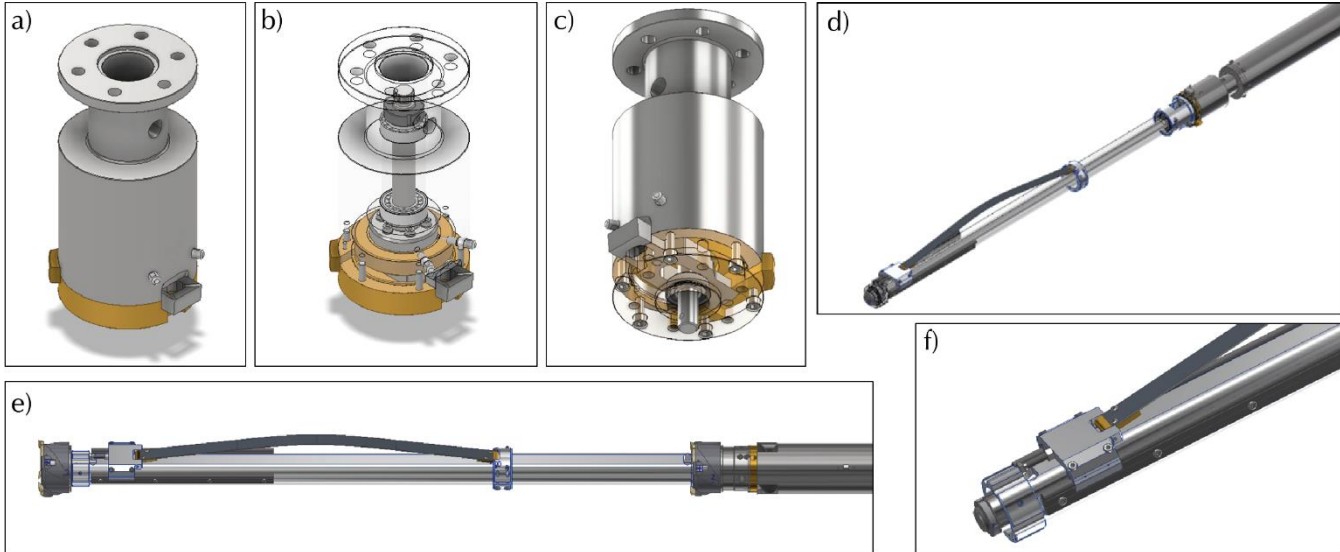

**Figure 4: a, b, c) The broaching tool in detail. d) The optional spring below the broaching tool. e,f) The milling head can be mounted either at the bottom or top of the spring sleeve, facing down (cut while descending) or facing up (cut while ascending). Only one milling head would be mounted at a time. The linear rail at the bottom of the spring can be limited in travel to produce lateral force, either by a bolt as shown above, or a spring for longer travel demands.**

## 2.4 Broaching

### 2.4.1 Design of the Broaching Tool

In the next step, we broach the downhill side of the borehole. The purpose of broaching is to make a groove in the borehole that the drill will use as an orientation reference; that is, the drill will follow this groove, pushing the cutters against the other side of the borehole wall while milling. A central shaft couples directly to the output of the drill motor section (fig. 4 and Supplement 1 - CIC-001868). The other end of the shaft is mated to a changeable cam-plate. The plate determines the overall depth-of-cut of the blade into the borehole wall. CIC-001871 (Supplement 2) is an example of a cam-plate that can cut a nominal 10 mm deep slot. Note that there is a 45° "deadband" of rotation when the blade is fully retracted and fully protruded. The motor position must be able to be controlled to engage the blade within that band.

After the blade is protruded, a groove is cut by pulling the system upwards with the winch. The blade was designed to evacuate the chips through a hole in the center of the blade (Supplement 3 - CIC-001874).



### 2.4.2 Planned Broaching Process

For the test, we broached in two steps; 2 mm and 5 mm deep. We used a cam-plate with a total cutting depth of 5 mm. Then by rotating the motor by 90 degrees, the first stage of the broaching tool is driven out of the housing and into the borehole wall for an approximate cutting depth of 2 mm (fig. 2b). Then we pulled the drill up 20 m. We lowered it again. By rotating another 90 degrees, we fully engaged the cutter of the broaching tool into the side of the borehole. By pulling up, we have now broached a 5 mm deep and 30 mm wide groove.

### 2.4.3 Execution, Issues, and Improvements

We broached the first step (2 mm depth) from 160 m to 139 m depth with a cable tension between 200 and 500 kg (yellow bar in fig. 3c, d). During the second broaching step (the full 5 mm) we reached cable tensions between 600 and 1400 kg (orange bar in fig. 3c, d). After five meters, the motor stopped functioning and we could not retract the blade. Therefore, we were forced to broach all the way to the surface. This was probably caused by the AT slide hammer's up-and-down motion during stick-slip, which broke the slip-ring connector. As the motor stopped functioning, the ascent is not recorded (upwards from approx. 154 m depth, fig. 3c, d). To avoid this technical issue in the future, we plan to include a mechanical fuse to release the broaching tool and make an ascent possible, even in the event of an electronic failure in the drill.

The broach blade was designed to evacuate chips through the center of the blade - conceptually similar to an ice screw. However, the chips that were collected following this step by the borehole filter revealed very densely packed pellets. This means the chips could not easily pass through the central channel. This should be improved for future iterations.

We can optionally mount the spring sleeve below the broach, with the spring fixed directly below the broaching blade (fig. 4d). When the spring drops into the groove, it will prevent the broach from cutting a helix into the borehole while pulling up. We did not rely on this mechanism for the rest reported here.

## 2.5 Milling

### 2.5.1 Design of the Milling Tool

Having cut a groove into the side of the borehole for orientation referencing, we lowered the second unit of the direction drilling tool down the hole: the spring sleeve (fig. 4e,f and Supplement 4 - CIC-001879) and milling head. The purpose of the spring sleeve is to generate a radial force, pushing the milling head into the side of the borehole. The milling head uses that force to cut sideways, eventually forming the new borehole. This technique was successfully used to create a replicate hole during the NEEM project (unpublished data). However, it was done in a dry borehole and relied solely on the force of gravity





to press the cutters into the borehole. The spring sleeve uses an AT blade as a spring (fig. 2c) which needs to sit in the broached slot. The spring is fixed to a collar on one end, and the other end rides on linear rails. The travel of the blade on the linear rail can be limited by a push-screw or spring to ensure there is adequate lateral force for milling.

### 2.5.2 Orientation of the Milling Tool

To position the spring into the slot, we rotated the drill backward. The one-directional bearings in the assembly guarantee that the spring rotates with the shaft of the motor. When we witnessed a peak in motor power consumption, we knew the spring had dropped into the broached slot (thus the AT blades need to rotate, which requires more power to slip). For verification, we rotated by 360 degrees and again witnessed the high motor power consumption. The power consumption we witnessed was very short and only visible on the direct readings of the power supply unit (Sørensen) and not recorded by the software; the

software subsamples over a couple of seconds, removing the peak in power consumption.

### 2.5.3 Configurations for Cutting

There are several positions and configurations for the milling head on the spring sleeve, as well as several mounting options for the spring sleeve to the drill. The milling head is a standard ice coring head, with full kerf cutters. This head can be mounted

at the top or the bottom of the spring sleeve, facing up or down (fig. 4). The spring sleeve can be mounted to the drill in three ways:
1.  At the end of the hollow shaft with a one-directional bearing.
2.  At the end of the core barrel with a one-directional bearing.
3.  Fixed to the chips chamber.


When cutting facing downwards we will create a sharp step for the deviation to begin, however, the chips will stay around the spring. When the blade faces up and the milling happens in the upward direction the chips will be left below the drill, reducing the risk of sticking the drill. For the test, we used only a downward-facing cutter at the bottom of the spring sleeve.

### 2.5.4 Execution

We performed the test of the directional drilling tool on May 31, 2024, between 14.25 h and 16.25 h (fig. 3b). For the cutting itself, we used two configurations of cutters, to test which one performed better:

First, we tested a milling head with only one cutter and a very slow decent speed (not plotted), We started the drill motor and pushed the mounted milling head into the side of the wall. We slowly lowered the drill, cutting into the side of the wall. Due

to very slow descending speeds and very little cutter force, the cutting itself was not witnessable on the drill's power





consumption. We milled five meters in total; two times while lowering the drill and one time while moving the drill upwards, pushing the cutters more and more into the side of the wall with each milling run.

The second milling process is color-coded in blue (fig. 3e, f) and was done between 15:15 and 15:45 on May 31, 2024. Here, we used a milling head with three normal cutters and "normal" drilling downward speed. In this case, we only milled 2 m on the side of the borehole but repeated this three times up and four times down (fig. 3b, milling). We stopped at the same depth as with the first setup, i.e. 145.10 m depth. The sharp drop in cable tension at around 15:25 or 143.6 m depth (fig. 3e or 3f, respectively) is due to a short power-off of the motor, yet with continuous cable payout.

We stopped every run at the same depth, this ensured that we cut out a ledge on the side of the borehole.

## 3 Proof of Concept – The Ledge Cut in the Side of the Borehole

After having milled the side of the borehole (around 15:45 in fig. 3f), we stopped the drill and continued to pay out cable. The cable tension sharply dropped, which indicated that we were resting the drill on the ledge we had just cut into the side of the borehole (brown curve in fig. 3g, h). We stopped at a cable tension of 63 kg, corresponding to almost 80 kg resting on the small ledge (fig. 3g).

We raised the drill by a few tens of centimeters, then turned the blade of the directional drilling tool, by 180 degrees, thus forcing it to come out of the broached slot. We slowly lowered the drill, and we did not catch the ledge, as we did not see a drop in load (fig. 3i, red curve). The spring was now pushing the milling head away from the ledge to the opposite side of the borehole.

For a final verification, we raised the drill approximately 30 cm above the ledge. We then rotated the spring by another 180 degrees and ensured that it caught the broaching slot again. We then lowered the drill and again caught the ledge, resulting in a drop of the cable tension (fig. 3j). The load did not drop as far as in the first test, as we probably broke off the small ledge by letting the drill rest on it.

This proves that we have managed to cut into the side of the borehole, and we have also managed to rest a large part of the drill's weight on this ledge. We did not perform any further directional drilling in the borehole, as the main goal was to prove the concept.

## 4 Discussion

### 4.1 Differences in our test compared to non-test deployment in the field

We tested the replicate drilling equipment as it is planned to be used in later projects (see introduction), however, the test was performed on the "uphill" side of the borehole rather than the "downhole" side. We broached on the "downhill" and cut into the "uphill" side of the borehole. For deviation/replicate drilling in the future the opposite should be done, to ensure the effect of gravity guides the drill into the new borehole (fig. 2d).

Furthermore, we also recommend broaching more than ten meters, as the long ice core drill needs to get the curvature of the new direction of the borehole. The best length of broaching still needs to be determined, which we could not do with our test.

For our test, we cut the ledge in the uppermost part of the borehole (140 m depth) to ensure short travel times while testing and to not disturb any bottom sections of the hole. This is not the standard procedure, and future project needs determine the depth. In our case, cutting in the upper part created light chips that floated to the top of the liquid level, creating a plug at the air-drill fluid interface. We drilled through this plug with ease, but lighter instruments could not penetrate it.

**5 Conclusion**

We introduced a new directional-drilling technique for the Danish HT drill, intended to be deployed for future deep ice-core projects at GRIP, Greenland, and Beyond EPICA (Little Dome C), Antarctica. We demonstrated that our tools and method performed in the manner we had designed them for by testing them in the EastGRIP deep borehole, Greenland. By cutting a ledge into the side of the EastGRIP borehole and resting the drill on it, we have proven that the concept works. Milling on the
side of which the drill will naturally rest due to gravity, will make it rather easy to use this setup and retrieve replicate ice cores of the sections of interest.

**Competing Interests**

The contact author has declared that none of the authors has any competing interests.

**Acknowledgments**

EGRIP is directed and organized by the Centre for Ice and Climate at the Niels Bohr Institute, University of Copenhagen. It is supported by funding agencies and institutions in Denmark (A. P. Møller Foundation, University of Copenhagen), USA (US National Science Foundation, Office of Polar Programs), Germany (Alfred Wegener Institute, Helmholtz Centre for Polar and Marine Research), Japan (National Institute of Polar Research and Arctic Challenge for Sustainability), Norway (University of Bergen and Trond Mohn Foundation), Switzerland (Swiss National Science Foundation), France (French Polar Institute
Paul-Emile Victor, Institute for Geosciences and Environmental research), Canada (University of Manitoba) and China (Chinese Academy of Sciences and Beijing Normal University).



**Data Availability**

The technical drawings and the borehole inclination data are in the supplement to this paper (URL:
https://erda.ku.dk/archives/27d80018e2b334947fd99cecbdc63e71/published-archive.html). The detailed drill log, i.e., the data
used to create figure 3, is available on request.

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
