# Peer review of "Brief Communication: The Danish Replicate Drilling System"

_EGUsphere, 2024_

## Referee Comment (RC1)

Brief Communication: The Danish Replicate Drilling System – Results from the First Field Test

Author(s): Westhoff and others

**General comments**
I find the paper is interesting and relevant, and worthy of publication. The paper is well illustrated, but could still be improved fairly easily.

First, I suggest inserting very early on a few lines (no more) on why a core deviation and/or a duplicate core might be needed. Incidentally, I would refer to the second core as a 'duplicate' and not a 'replicate'.

Second, I would consider the method-related terminology and stick to it throughout; as written, I have no doubt that the terms are correct, but I was still a little confused by the process (involving milling, cutting, grooving and broaching). I think this comes from a need for a simple (or at least as simple as possible given it is quite mechanically technical) explanation of the process from the outset, followed by consistent use. If I follow the technique correctly, I might suggest a summary something like: "The method is based on adapting the corer to incorporate three key functions. First, a retractable broaching tool cuts a vertical groove, ~30 mm wide and up to * mm deep, along the borehole wall. Second, a spring sleeve, which bows into and slides along that groove, retains the corer in a consistent and known orientation. This spring sleeve also pushes the base of the corer laterally away from the keyway, raising the cutting head's contact force on the opposite side of the borehole wall. Third, a milling head (with the ability to cut sideways as well as downwards) is used to mill into the opposite side of the borehole wall under this enhanced force." (Incidentally, one could also refer to the groove as a 'keyway', but I'm not sure the technical accuracy outweighs the rarity of the term; 'groove' would be good enough for me).

Third, and again if I understand correctly, the reported application demonstrates the use of the technique to create a shelf from which it should be straightforward to core a new hole. However, this duplicate coring is not guaranteed, and the manuscript does not actually report that new duplicated coring. This needs to be acknowledged.

Fourth, several pointers for future improvements, refinements and applications are given at various places in the manuscript – most notably in 2.4.3. However, this is not the only place potential improvements are raised or implied. I would retitle 2.4.3 as 'Trial application' and insert a new subsection on 'Future improvements' (or similar) into the Discussion or the Conclusions.

**Specific comments**

| Line/Location | Comment/Suggestion |
|---|---|
| | |
| 11 | '…in the EastGRIP…' |
| 11- | Here, I think the explanation would benefit from being presented more clearly. It also doesn't have to be the downhill side – in fact, I think this is a bit of a red herring and I might not mention it here at all. Perhaps mention that the process can be assisted by using gravity on a non-vertical section of borehole. |
| 20 | I'd delete '…in the borehole…' to end of sentence. |
| 24 | 'The replicate system…' (and I'd refer to it consistently and solely as a 'duplicate' system). Given the two possible uses, I might even refer to it as a 'deviating/duplication system' (sounds awful though). |
| 26 | '…2500 m…' (insert space) |
| 38-41 | 'We performed another test at the NEEM site, in a dry 400 m deep borehole of local inclination ~4°. Here, the inclination was sufficient to mill into the side of the borehole under gravity alone, cutting a quasi-horizontal ledge into the borehole wall.' (I'd not dwell on this being unpublished, since you are doing so here). |

| | |
|---|---|
| 56 & 105-110 | The manuscript presents little information on this orientation package and the data are also not presented. I think this verification claim does need to be demonstrated in the main text. Alternatively, if the data somehow fall short, at least alternatively refer to 'future application of... But it'll still ideally need a reference, and it would be nice to see the corroborating inclination data. |
| 60-61 | I would put the source attribution in the Figure caption and just refer to 'Supplement 8' at the appropriate point in the text. |
| Fig 2 (& 71) | I'm not convinced this needs to be inclined. Since the technique needs to be deployable anywhere along a borehole (by the manuscript's own requirement section) then a preexisting inclination cannot be a requisite. Also, for me it detracts from the core technique of the spring pushing from the keyway. It also confuses since the application in the manuscript is the other way around… I'd just mention that a pre-existing inclination helps mill into the downhill side (with the uphill side broached) – as long as the orientation is suitable for the need. Perhaps all of 71 – 75 can be reworded to account for this.

I would label spring sleeve and ledge on panel C; also the groove/keyway on B. |
| 76 | 'The system needs to comply with certain operational requirements:' |
| 83 | '…diameter.' |
| 84 | The system must operate at… |
| 2.2 Subtitle | 'System deployment and testing' (?) |
| Fig 3 | Are there any orientation data to refine panels H-J? |
| 90 | '..cable tension excursions.' 'g-j) show lateral milling of a ledge in the borehole wall, indicated by…' |
| 105 – 110 | Are these orientation data not available to be shown as a log alongside e.g. Fig 3H-J? See also comment on line 56 above. |
| 120 | '..groove in the borehole wall.' |
| 120 - 123 | Again, this is a slightly different way of describing the technique and process. I would select one description and either not repeat or, if repetition is needed, stick to almost exactly the same wording to avoid any possible confusion. I would also remove the role of off-vertical inclination from the primary description – in the first instance assuming a vertical borehole and only once described noting that an off vertical inclination can help through gravity. |
| 133-134 | We then raised the drill by 20 m and lowered it again. By rotating another 90° (resulting in a total rotation of 180°)… |
| 134-135 | Can the 5 mm deep keyway be explained? Is it that a certain depth of material is removed during each pass? |
| 148 | Move to new future refinements section? |
| 151 | Interesting. Just from personal reference, I imaged what I think must have been a similar helix (I imagine from the normal teeth) at ~170 m depth in the NEEM borehole wall. See Figure 3d here: http://dx.doi.org/10.3189/2013aog64a201. Happy to share the original if you want it – but I don't think this paper needs it. |
| 157 | 'The spring sleeve is designed to push the milling head into the opposite side of the borehole wall.' (This is simpler and avoids reference to a 'radial' force – which I am not confident of). |
| 161 | 'AT' needs defining |
| 170 | Sampling frequency improvement could be included as a future refinement. |
| 182 | 'During upwards drilling, the blade faces upwards and the chips…' |
| 188 | '…(not plotted). We started…' |
| 190 | '…slow descent and…' |
| 196-197 | I leave this up to the authors, but I think I would remove the effects of this power-outage from the data (and note that it was done); it is clearly an artefact. |
| 200 | 'After milling into the borehole wall…' |
| 205 | '…the ledge, as evidenced by no drop in…' |
| 210 | '…test, possibly by degrading the integrity of the ledge by repeated contact.' |

| | |
|---|---|
| 217-220 | I'm not sure this distinction needs spelling out again – the manuscript already stated that the test was the 'wrong way around'. |
| 224 | A future development to add to the list? Delete '…, which we could not do with our test' |
| 225 | Also need to consider chip removal as a future development since one of the manuscripts stated requirements is to be able to deviate-duplicate at any depth (below casing I imagine). |
| 235 | '… will improve further the effectiveness of this technique by supplementing the force imparted by the spring sleeve with that resulting from gravity.' |

---

## Referee Comment (RC2)

Brief Communication: The Danish Replicate Drilling System – Results from the First Field Test

Authors: Westhoff and others

**General Comments**

The concept and method presented in this paper for creating a notch in the wall of an ice borehole are novel and unique and worth publication. Overall, the content was well written and supported by the figures.

Was there a reason the inclination plots were not included as evidence of success in this paper? It is an important and critical feedback that can be used to determine the depth of both the broached groove and milled notch. Cable tension alone only doesn't give a good indication of how successful the milling operation was or if a large enough notch has been created to move on to the next step. Was a borehole camera deployed to get video or pictures of the groove or milled notch/step? If so, the images would be very interesting to include in this paper.

I recommend adding a section before the conclusion describing the next steps, modifications, and plans for further testing to demonstrate a full deviation can be completed and replicate cores recovered using this method.

**Specific Comments**

Lines 11-14: The sentence beginning with "By determining the borehole orientation…" does not make much sense as written and could use rewording for better clarity. I feel the following sentence misguides the reader into thinking that the purpose of the ledge is just for setting the weight of the drill on where I think the significance of the milling is to create a new guiding path for the core drill to exit the parent borehole. The two sentences could be replaced with something like "A groove is first cut on the uphill side of the borehole wall using a broaching process. This groove is then used to guide a milling tool to produce a circular notch and ledge in the downhill side of the borehole. Gravity would now guide the ice core drill into this newly formed notch diverging from the parent borehole, gradually producing full diameter replicate ice cores."

Section 1: I suggest adding a few sentences at the beginning of this section describing the benefits and importance of replicate coring and why it is important to continue to develop this technology.

Figure 2: Panels C & D show the new/duplicate hole inclined beyond vertical. This may give some readers the wrong impression and I suggest editing the schematic, so the new/duplicate borehole isn't shown inclined beyond vertical. I think it should be mentioned somewhere in the paper that the parent borehole must have a certain amount of inclination and the new/duplicate borehole must have an inclination between 0 and less than that of the parent borehole for this technique to work as presented. I also recommend labeling the key parts (broaching cutter, groove, spring, and milling head) in the pictures.

Line 80: Doesn't the parent borehole need to have inclination greater than 0 for this to work?

Figure 4: Labeling or highlighting in a bright color the key parts (broaching cutter, milling head, spring, and linear slide and bolt for limiting the travel) in the pictures that are described in the figure text would be helpful.

Lines 133 to 135: I suggest the rewording for better readability. "The drill is then pulled up 20 m to complete the first 2 mm deep cut. After lowering the drill back to the starting depth, the motor is rotated another 90 degrees, extending the broaching tool the full 5 mm into the side of the borehole. The drill is once again raised 20 m to complete the second cut, leaving a 5 mm deep and 30 mm wide groove."

Line 141: Change this first instance of "AT" to "Anti-Torque (AT)".

---

## Author Comment (AC1)

**Reply to review #1**

Brief Communication: The Danish Replicate Drilling System – Results from the First Field Test

Author(s): Westhoff and others

**General comments**

I find the paper is interesting and relevant, and worthy of publication. The paper is well illustrated, but could still be improved fairly easily.

Thank you very much for the positive statement.

First, I suggest inserting very early on a few lines (no more) on why a core deviation and/or a duplicate core might be needed. Incidentally, I would refer to the second core as a 'duplicate' and not a 'replicate'.

We will include a few lines in the beginning introducing the necessity a bit more. We will also change the terminology to duplicate.

Second, I would consider the method-related terminology and stick to it throughout; as written, I have no doubt that the terms are correct, but I was still a little confused by the process (involving milling, cutting, grooving and broaching). I think this comes from a need for a simple (or at least as simple as possible given it is quite mechanically technical) explanation of the process from the outset, followed by consistent use. If I follow the technique correctly, I might suggest a summary something like: "The method is based on adapting the corer to incorporate three key functions. First, a retractable broaching tool cuts a vertical groove, ~30 mm wide and up to * mm deep, along the borehole wall. Second, a spring sleeve, which bows into and slides along that groove, retains the corer in a consistent and known orientation. This spring sleeve also pushes the base of the corer laterally away from the keyway, raising the cutting head's contact force on the opposite side of the borehole wall. Third, a milling head (with the ability to cut sideways as well as downwards) is used to mill into the opposite side of the borehole wall under this enhanced force." (Incidentally, one could also refer to the groove as a 'keyway', but I'm not sure the technical accuracy outweighs the rarity of the term; 'groove' would be good enough for me).

Thank you very much for the suggestions. We will implement the explanation given by you to increase clarity. We will add this to a short section where we introduce the terminology.

Third, and again if I understand correctly, the reported application demonstrates the use of the technique to create a shelf from which it should be straightforward to core a new hole. However, this duplicate coring is not guaranteed, and the manuscript does not actually report that new duplicated coring. This needs to be acknowledged.

We agree that a duplicate core is not drilled in our test. The shelf nevertheless is a very high guarantee to produce this core, as has been demonstrated, e.g. in the NEEM core. We will acknowledge this and clarify.

Fourth, several pointers for future improvements, refinements and applications are given at various places in the manuscript – most notably in 2.4.3. However, this is not the only place potential improvements are raised or implied. I would retitle 2.4.3 as 'Trial application' and insert a new subsection on 'Future improvements' (or similar) into the Discussion or the Conclusions.

We will include a section about further improvements to the end of the manuscript as suggested.

| Specific comments Line/Location | Comment/Suggestion |
|---|---|
| 11 | '…in the EastGRIP…' will be corrected. |
| 11- | Here, I think the explanation would benefit from being presented more clearly. It also doesn't have to be the downhill side – in fact, I think this is a bit of a red herring and I might not mention it here at all. Perhaps mention that the process can be assisted by using gravity on a non-vertical section of borehole. Thank you for the suggestion. We will adjust it accordingly. |
| 20 | I'd delete '…in the borehole…' to end of sentence. Will be done. |
| 24 | 'The replicate system…' (and I'd refer to it consistently and solely as a 'duplicate' system). Given the two possible uses, I might even refer to it as a 'deviating/duplication system' (sounds awful though). |
| | According to EPA.gov |
| | "**Duplicate**: an adjective describing the taking of a second sample or performance of a second measurement or determination. Often incorrectly used as a noun and substituted for "duplicate sample." Replicate is to be used if there are more than two items. See **Replicate**. |
| | **Replicate**: an adjective or verb referring to the taking of more than one sample or to the performance of more than one analysis. Incorrectly used as a noun in place of replicate analysis. Replicate is to be used when referring to more than two items. See **Duplicate**." |
| | For the first deviation from the bore hole, we thus drill a duplicate core. Yet we would like to leave the option of also making a 3rd hole further up to get another sample. This would make it a replicate tool. We would therefore prefer the term Replicate. |
| 26 | '…2500 m…' (insert space) Will be done. |
| 38-41 | 'We performed another test at the NEEM site, in a dry 400 m deep borehole of local inclination ~4°. Here, the inclination was sufficient to mill into the side of the borehole under gravity alone, cutting a quasi-horizontal ledge into the borehole wall.' (I'd not dwell on this being unpublished, since you are doing so here). Thanks for the suggestion, we will adapt this. |
| 56 & 105-110 | The manuscript presents little information on this orientation package and the data are also not presented. I think this verification claim does need to be demonstrated in the main text. Alternatively, if the data somehow fall short, at least alternatively refer to 'future application of… |

But it'll still ideally need a reference, and it would be nice to see the corroborating inclination data. See comment to figure 3.

| | |
|---|---|
| 60-61 | I would put the source attribution in the Figure caption and just refer to 'Supplement 8' at the appropriate point in the text. Will be changed |
| Fig 2 (& 71) | I'm not convinced this needs to be inclined. Since the technique needs to be deployable anywhere along a borehole (by the manuscript's own requirement section) then a preexisting inclination cannot be a requisite. Also, for me it detracts from the core technique of the spring pushing from the keyway. It also confuses since the application in the manuscript is the other way around... I'd just mention that a pre-existing inclination helps mill into the downhill side (with the uphill side broached) – as long as the orientation is suitable for the need. Perhaps all of 71 – 75 can be reworded to account for this. |
| | There is actually a need for some inclination. Yes, we can mill in a plumb hole because we have the spring. But the drill will not fall onto that ledge if there is no inclination, it will stay in the original bore hole. Therefore, it's true that we do not meet the requirement of deploying anywhere in the borehole with this requirement. However in our experience, there has never been a perfectly plumb hole, so we accepted this short-coming for the technique. |
| | I would label spring sleeve and ledge on panel C; also the groove/keyway on B. |
| | Will be added |
| 76 | 'The system needs to comply with certain operational requirements:' Thanks for the suggestion |
| 83 | '...diameter.' |
| 84 | The system must operate at... |
| 2.2 Subtitle | 'System deployment and testing' (?) Thanks for the suggestion, we will adjust it. |
| Fig 3 | Are there any orientation data to refine panels H-J? |
| | To verify the rotation of the drill, we used the live-rotation in a few-degree increments. This does not change the orientation of the drill's azimuth and inclination and is not recorded in our software. |
| 90 | '..cable tension excursions.' 'g-j) show lateral milling of a ledge in the borehole wall, indicated by...' Will be adjusted |
| 105 – 110 | Are these orientation data not available to be shown as a log alongside e.g. Fig 3H-J? See also comment on line 56 above. |
| 120 | '..groove in the borehole wall.' Will be adjusted |
| 120 - 123 | Again, this is a slightly different way of describing the technique and process. I would select one description and either not repeat or, if repetition is needed, stick to almost exactly the same wording to avoid any possible confusion. I would also remove the role of off-vertical inclination |

from the primary description – in the first instance assuming a vertical borehole and only once described noting that an off vertical inclination can help through gravity.

Thanks for the suggestion, we will adjust it.

133-134     We then raised the drill by 20 m and lowered it again. By rotating another 90° (resulting in a total rotation of 180°)... Will be added

134-135     Can the 5 mm deep keyway be explained? Is it that a certain depth of material is removed during each pass?

We will add the following: A 5-mm groove I sufficient and necessary to guide the spring of the milling tool for the next step.

148     Move to new future refinements section? Thanks for the suggestion, we move this statement.

151     Interesting. Just from personal reference, I imaged what I think must have been a similar helix (I imagine from the normal teeth) at ~170 m depth in the NEEM borehole wall. See Figure 3d here: http://dx.doi.org/10.3189/2013aog64a201. Happy to share the original if you want it – but I don't think this paper needs it. Thanks for hinting us to the paper, we will reference it for visualization

157     'The spring sleeve is designed to push the milling head into the opposite side of the borehole wall.' (This is simpler and avoids reference to a 'radial' force – which I am not confident of).

Thanks for the suggestion, we will adapt to your phrasing.

161     'AT' needs defining

170     Sampling frequency improvement could be included as a future refinement. That is a good point, thank you.

182     'During upwards drilling, the blade faces upwards and the chips...' Thanks for the suggestion

188     '...(not plotted). We started...'

190     '...slow descent and...'

196-197     I leave this up to the authors, but I think I would remove the effects of this power-outage from the data (and note that it was done); it is clearly an artefact. Thanks for the suggestion, we will remove the artefact datapoints and mention it.

200     'After milling into the borehole wall...'

205     '...the ledge, as evidenced by no drop in...'

210     '...test, possibly by degrading the integrity of the ledge by repeated contact.'

217-220     I'm not sure this distinction needs spelling out again – the manuscript already stated that the test was the 'wrong way around'. We will remove this section.

224     A future development to add to the list? Delete '..., which we could not do with our test'

225     Also need to consider chip removal as a future development since one of the manuscripts stated requirements is to be able to deviate-duplicate at any depth (below casing I imagine). We will include the chips removal to future development. Below the casing is stated in line 85.

235     '… will improve further the effectiveness of this technique by supplementing the force imparted by the spring sleeve with that resulting from gravity.'

Thanks for the suggestion and all the detailed comments to improve the quality of the manuscript.

---

## Author Comment (AC2)

**Reply to review #2**

Brief Communication: The Danish Replicate Drilling System – Results from the First Field Test

Authors: Westhoff and others

**General Comments**

The concept and method presented in this paper for creating a notch in the wall of an ice borehole are novel and unique and worth publication. Overall, the content was well written and supported by the figures.

Thank you very much for the positive statement and the review.

Was there a reason the inclination plots were not included as evidence of success in this paper? It is an important and critical feedback that can be used to determine the depth of both the broached groove and milled notch. Cable tension alone only doesn't give a good indication of how successful the milling operation was or if a large enough notch has been created to move on to the next step. Was a borehole camera deployed to get video or pictures of the groove or milled notch/step? If so, the images would be very interesting to include in this paper.

To verify the rotation of the drill, we used the live-rotation in a few-degree increments. This does not change the orientation of the drill's azimuth and inclination and is not recorded in our software.

We attempted to use a borehole camera, but the fiber-optic cable camera did not manage to acquire images. The high-pressure sealed GoPro captured images, but due to the very cloudy liquid these were not usable.

I recommend adding a section before the conclusion describing the next steps, modifications, and plans for further testing to demonstrate a full deviation can be completed and replicate cores recovered using this method.

Thanks for the recommendation. We will add this section before the conclusion.

**Specific Comments**

Lines 11-14: The sentence beginning with "By determining the borehole orientation…" does not make much sense as written and could use rewording for better clarity. I feel the following sentence misguides the reader into thinking that the purpose of the ledge is just for setting the weight of the drill on where I think the significance of the milling is to create a new guiding path for the core drill to exit the parent borehole. The two sentences could be replaced with something like "A groove is first cut on the uphill side of the borehole wall using a broaching process. This groove is then used to guide a milling tool to produce a circular notch and ledge in the downhill side of the borehole. Gravity would now guide the ice core drill into this newly formed notch diverging from the parent borehole, gradually producing full diameter replicate ice cores."

Thank you very much for the suggestion. We will rephrase the sentences as suggested.

Section 1: I suggest adding a few sentences at the beginning of this section describing the benefits and importance of replicate coring and why it is important to continue to develop this technology. Thanks for the suggestion, we will add this.

Figure 2: Panels C & D show the new/duplicate hole inclined beyond vertical. This may give some readers the wrong impression and I suggest editing the schematic, so the new/duplicate borehole isn't shown inclined beyond vertical. I think it should be mentioned somewhere in the paper that the parent borehole must have a certain amount of inclination and the new/duplicate borehole must have an inclination between 0 and less than that of the parent borehole for this technique to work as presented. I also recommend labeling the key parts (broaching cutter, groove, spring, and milling head) in the pictures.

Thanks for the suggestion. We will adapt the figure with the vertical orientation and the labeling of the key parts. We will also elaborate on the new borehole's inclination

Line 80: Doesn't the parent borehole need to have inclination greater than 0 for this to work?

We would like to have an inclination of 2° (or more) for the replicate operations. While we can start the deviation process in a plumb hole, later steps require that the drill "falls" into the new hole - requiring some inclination.

Figure 4: Labeling or highlighting in a bright color the key parts (broaching cutter, milling head, spring, and linear slide and bolt for limiting the travel) in the pictures that are described in the figure text would be helpful.

Thanks for the suggestion, we will add the labels.

Lines 133 to 135: I suggest the rewording for better readability. "The drill is then pulled up 20 m to complete the first 2 mm deep cut. After lowering the drill back to the starting depth, the motor is rotated another 90 degrees, extending the broaching tool the full 5 mm into the side of the borehole. The drill is once again raised 20 m to complete the second cut, leaving a 5 mm deep and 30 mm wide groove." Thank you for the suggestion, we will implement it and combine it with the suggestion from reviewer #1, who also suggested changing this section.

Line 141: Change this first instance of "AT" to "Anti-Torque (AT)". Will be done.

---

## Author Response (AR1)

Dear editor and reviewers,

We have accommodated all the comments and suggestions from the reviewers, which were stated in the individual discussions. Please find the track-changes version and the new version of the manuscript uploaded here.

Thank you for the reviews.

All the best,

Julien Westhoff et al.